

# A study of female tennis players: Speedcourt training is effective on improving agility and change-of-direction

Zhihui Zhou[1,2,*], Jiawei Wang[3,*], Hao Wang[2], Guo Ru[4] and Fanhui Kong[5]

[1] Anhui Science and Technology University, Anhui, China
[2] Beijing Sport University, Beijing, China
[3] Beijing Normal University, Beijing, China
[4] Wenzhou Business College, Zhejiang, China
[5] Fudan University, Shanghai, China
[*] These authors contributed equally to this work.

## ABSTRACT

**Objective**. The aim of the present paper was to determine the impact of Speedcourt training on agility and change-of-direction (COD) in female tennis players, and to research the relevance between agility and lower limbs unilateral explosive power (UEP). Despite extensive research on agility training, limited studies have explored these effects specifically in female athletes and the number of exercises such as Speedcourt is also small, necessitating this investigation.

**Method**. Twenty-two female tennis players underwent SpeedCourt training for 6 weeks, respectively executed random sequence shuttle run training (RS group, $N = 11$, age: $22.36 \pm 1.21$ years) and fixed sequence shuttle run training (FS group, $N = 11$, age: $22.27 \pm 1.27$ years). The spider run, T-drill, reactive agility (RA) and triple crossover hop (TCH) before and after intervention were measured. And the TCH tested the left and right legs separately to detect the subject's UEP.

**Results**. The two-way repeated measures analysis of variance showed significant improvements in spider run ($p < 0.001$, partial $\eta2 = 0.95$), T-drill ($p < 0.001$, partial $\eta2 = 0.94$) and RA ($p < 0.001$, partial $\eta2 = 0.96$). The RS group demonstrated significantly greater improvements in RA compared to the FS group, with statistical significance ($p < 0.05$, partial $\eta2 = 0.184$). And agility related tests showed moderate to strong correlations with unilateral explosive power.

**Conclusion**. Six-week Speedcourt training can effectively enhance the agility and change-of-direction of female tennis players. Incorporating lower limb explosive exercises into agility-specific training may further enhance agility improvements in female tennis players.

Corresponding authors
Guo Ru, xinchenxi@sufe.edu.cn
Fanhui Kong,
2021241020@bsu.edu.cn

# INTRODUCTION

Tennis is a high-intensity, intermittent sport that demands a combination of sport-specific technical proficiency and a high degree of various physical components (*Fernandez-Fernandez et al., 2009*). Agility is a complex composite ability, which encompasses the following aspects: the capacity to swiftly alter body position and direction during movement, the ability to rapidly transition between actions, and the central nervous system's capability to quickly respond to external stimuli, which is to say, the ability to change direction and reactive agility (*Zhao, Ge & Sun, 2012*). Tennis requires players to perform many change-of-direction (COD) over the course of a match (*Giles, Peeling & Reid, 2021*). At a high level of tennis, players make about 2–4 COD per rally (*Murias et al., 2007*; *Fernandez-Fernandez et al., 2007*), and more than 100 COD per set (*Giles, Peeling & Reid, 2024*). At the same time, most of the rallies were performed around the baseline, and lateral and multi-directional moves occurred more often than straight moves (*Pereira et al., 2017*), thus suggesting that players often make sharp COD to return to the center of the court after hitting the ball back to the opposing side (*Palut & Zanone, 2007*). In other words, rapidly stop, go, and COD speed constitute the major performance determinants in tennis (*Fernandez-Fernandez, Ulbricht & Ferrauti, 2014*). The ability to COD is characterized by a change in the direction of movement that is already known in advance, it is planned, and the player does not need to respond to a certain stimulus (*Munivrana, Jelaska & Tomljanović, 2022*). In tennis, however, players must exhibit COD in response to external stimuli, such as the ball's trajectory, the opponent's movements, and other court dynamics. These instances of COD are typically reactive and not pre-planned (*Donoghue & Ingram, 2001*; *Cooke & Davey, 2005*). The capacity to swiftly respond to external stimuli with suitable actions or reactions during both training and competition is referred to as reactive agility (*Sheppard & Young, 2006*). Reactive agility (RA) includes cognitive processing, observation skills, and decision-making factors (*Sekulic et al., 2017*). Studies have shown that COD and RA alone represent different motor abilities (*Čoh et al., 2018*), and COD cannot be directly equated with RA, but are completely independent abilities (*Young, Dawson & Henry, 2015*; *Nimphius et al., 2017*).

The Speedcourt instrument is designed for testing and training speed, agility, coordination, COD and cognition, and has the most advanced speed-sensitive training programs. It can design different tests and training programs for different sports (*Li & Ding, 2021*). The reliability and effectiveness of the Speedcourt have been demonstrated (*Düking, Born & Sperlich, 2016*), and it was determined that the Speedcourt training regimen was more effective than conventional methods in enhancing COD and RA abilities (*Born et al., 2016*). Studies have shown that Speedcourt has achieved good results in improving the COD and RA of male collegiate tennis players (*Zhou et al., 2024*). A research reported that female players made COD more than 400 times per match during the Australian Open (2016–2018) (*Giles, Peeling & Reid, 2024*). The abilities to COD and RA are also crucial for female tennis players. However, in the existing research, compared with the male group, there are few studies on female tennis players (*Pluim et al., 2023*; *Lisi & Grigoletto, 2021*; *Carboch et al., 2019*). Current research exhibits a pronounced gender disparity in

investigating COD and agility development, with predominant focus on male tennis athletes and a paucity of studies addressing female counterparts (*Morais et al., 2024a*; *Lopez-Samanes et al., 2021*; *Sinkovic et al., 2023*; *Morais et al., 2024b*). Furthermore, while empirical evidence has established the efficacy of Speedcourt training systems in enhancing COD and agility among teamwork and badmiton athletes (*Li & Ding, 2021*; *Raeder et al., 2024*), its specific adaptation and impacts on female tennis players remain scientifically underexplored. Studies have shown that unilateral explosive power (UEP) is an important factor affecting agility and COD in tennis players, but the correlation between the three is controversial (*Munivrana, Filipčić & Filipčić, 2015*; *Hernández-Davó et al., 2021*). We hypothesize that Speedcourt training will significantly improve COD and RA, with the RS group exhibiting greater improvements compared to the FS group. Therefore, the aim of this study is to assess the efficacy of Speedcourt training in enhancing agility (COD and RA) among female tennis players, as well as to explore the correlation between agility (COD and RA) and UEP.

## MATERIALS AND METHODS

### Participants

Referencing prior research (*Zhou et al., 2024*), 22 female tennis players were randomly allocated to either the random sequence shuttle run training group (RS group, $N = 11$, mean age: $22.36 \pm 1.21$ years, mean training experience: $6.64 \pm 0.64$ years) or the fixed sequence shuttle run training group (FS group, $N = 11$, mean age: $22.27 \pm 1.27$ years, mean training experience: $6.86 \pm 0.60$ years) using a random number table. Participant details are presented in Table 1. The inclusion criteria were as follows: (1) absence of lower limb injuries; (2) no history of medication or surgery within the past 6 months; (3) training experience ranging from 6 to 8 years; (4) an International Tennis Number (ITN) score of 5 or higher; (5) a minimum attendance rate of 95%; (6) participants hadn't previous agility-specific training. Participants were instructed to maintain their usual lifestyle and physical activity levels, and to refrain from participating in any other experiments during the intervention. Written informed consent was obtained from all participants prior to the study. Ethical approval for this research was granted by the Ethics Committee of Beijing Sport University (2024202H).

### Procedures

#### Testing procedure

The testing procedures were conducted rigorously both pre-intervention and post-intervention. Before each test, participants engaged in a normalized warm-up (10 min: 5 min' jogging +5 min' dynamic stretching, the intensity reached grade 4–5 of RPE scale). After the warm-up, start the formal test with an interval of about 3 min.

The reliability and validity of T-drill test (Fig. 1) have been established in a previous study (*Pauole et al., 2000*). The Smartspeed photoelectric gate (Australia) was used for testing. The photoelectric gate was adjusted to the single turn-back time (point A). The athlete stood at point A, moved to point B (about 5.5 m) and touched the bottom of the B cone barrel with his right hand, and moved to the left side and touched the bottom of the

**Table 1  Basic information of subjects.**

| Group | N | Age (years) | Height (cm) | Weight (kg) | ITN (years) | Training experience (years) |
|---|---|---|---|---|---|---|
| RS | 11 | 22.36 ± 1.21 | 166.91 ± 3.01 | 59.36 ± 3.30 | 4.09 ± 0.70 | 6.64 ± 0.64 |
| FS | 11 | 22.27 ± 1.27 | 166.18 ± 2.99 | 57.73 ± 3.55 | 4.00 ± 0.78 | 6.86 ± 0.60 |

Notes.
FS, fixed sequence; RS, random sequence; ITN, international tennis number.

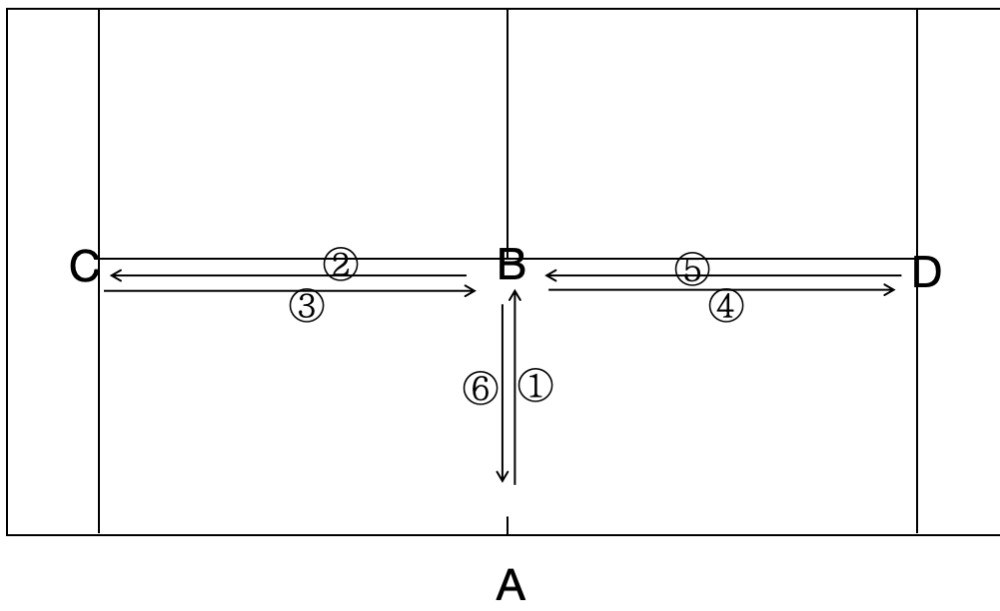

**Figure 1  T-drill test route.**

C cone barrel with his left hand (about 4.1 m). After that, slide 8.2 m to the right and touch the bottom of the D cone with your right hand. Then slide 4.1 m to the left and touch the bottom of the B cone with your left hand. Finally, the athlete backs off and crosses cone barrel A at the finish line. Take the best score in three tests, accurate to 0.01 s.

Previous studies have demonstrated that the spider run test is effective for evaluating tennis-specific COD speed (*Department of Youth Sports of China, 2012*). The test player is at the middle point of the court bottom line, and place a racket at the middle point, based on the tennis bottom line to the service line, the two sides of the tennis court singles line as the boundary, in accordance with the direction of the counterclockwise position on each T point. At the beginning of the test, the test players took the ball back to the racket one by one, and recorded the time spent in the whole return process. The test score was the time when the last ball was put into the racket. In case of running wrong route, dropping the ball, slipping, *etc.*, during the test, the test was repeated once. Each participant underwent

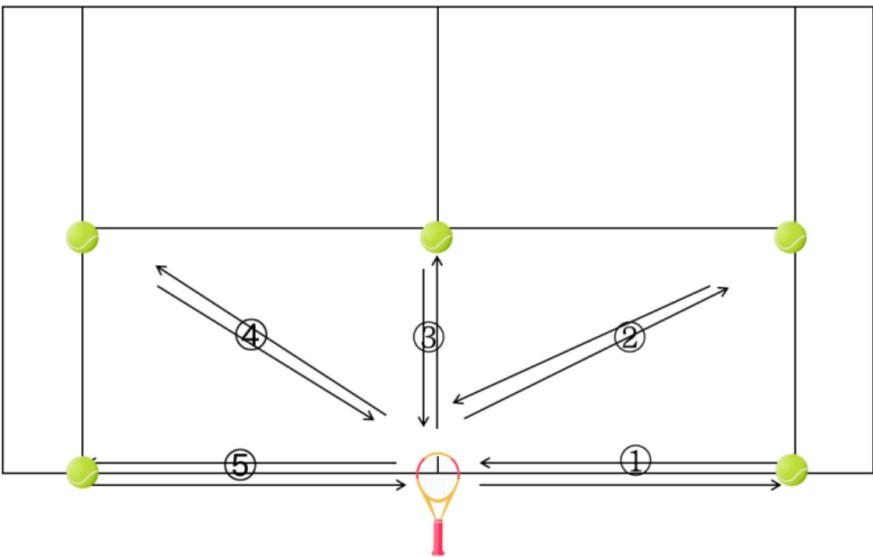

**Figure 2 Spider run test route.**

three attempts, with only the highest score being recorded for analysis. The layout of the test is illustrated in Fig. 2.

The RA test had been demonstrated good to excellent reliability (*Büchel et al., 2022*). And the test was conducted on the Speedcourt in this study. The layout of the test is illustrated in Fig. 3. Point 5 is the start and end point of each repeat sprint, and the sprint is carried out according to the random green light signal on the screen. Each green signal was accompanied by the interference signal of yellow light. The testing protocol commences when the participant initiates movement from Point 5 and terminates upon their return to the same point following completion of shuttle sprints traversing all eight remaining designated markers. The total reaction time was recorded by Speedcourt system. The experimental protocol required participants to complete three maximal-effort trials and retained the highest score. After each sprint, participants had a rest period of more than 30 s to ensure complete recovery before starting the next sprint at full capacity.

The triple crossover hop (TCH) for distance test was utilized to measure unilateral explosive power (UEP) (*Sagat et al., 2023*; *Chmielewski et al., 2024*). The layout of the test is illustrated in Fig. 4. Participants were required to execute three successive hops to cover the maximum distance possible in a forward direction (all on one limb, without pauses between hops except for the final landing), while crossing over a custom-made mat that is 15 cm wide, ensuring not to touch the mat. A trial was deemed successful if the participant landed controllably on the last hop. Each leg's test was conducted three times, with a 1-minute rest interval between attempts.

### Training procedure

All training sessions in this study were conducted using the Speedcourt system (Globalspeed GmbH, Hemsbach, Germany). This apparatus consists of a computer connected to a TV

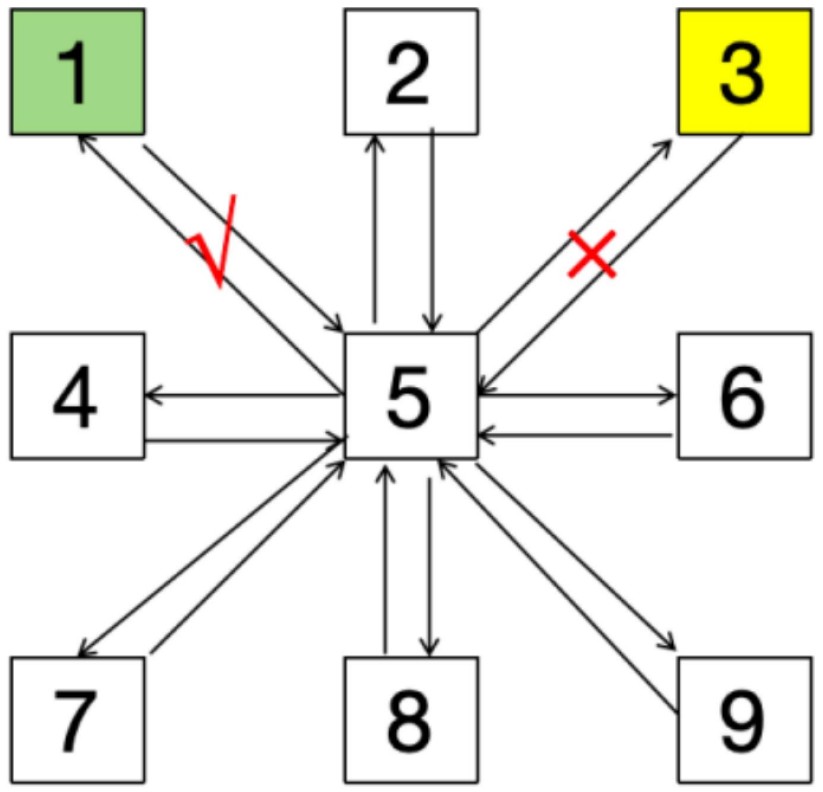

**Figure 3** Reactive agility test.

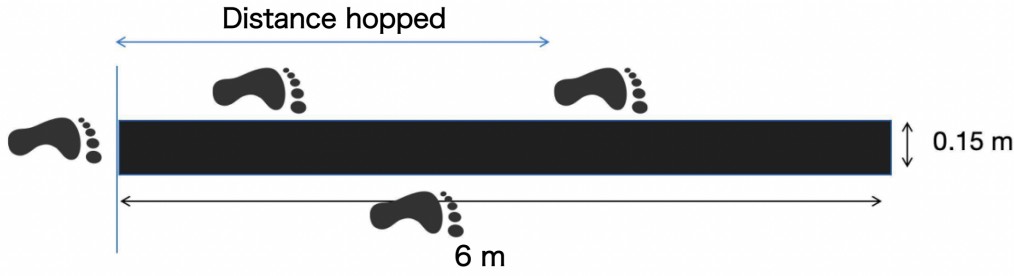

**Figure 4** Triple crossover hop test.

screen and radar sensors positioned beneath the device to detect a 3-by-3 grid. The grid is composed of squares, each measuring 40-by-40 cm, evenly spaced across a 5.5 m × 5.5 m court. Participants view a digital depiction of the court on the display. During the test, individual squares illuminate on the screen, indicating to the participant where to run next. Upon touching the lit square, another square lights up on the screen, signaling the next destination.

**Table 2  Training plan.**

| Training week | Exercise | Sets × reps | Pause(set/rep) |
|---|---|---|---|
| 1–2 | FS/RS | 3 × 5 | 5 min/30 s |
| 3–4 | FS/RS | 4 × 5 | 5 min/30 s |
| 5–6 | FS/RS | 5 × 5 | 5 min/30 s |

Notes.

FS, fixed sequence shuttle run training; RS, random sequence shuttle run training.

Existing evidence demonstrates that a six-week targeted agility training intervention elicits statistically significant improvements in athletes' COD and agility performance (*Thieschäfer & Büsch, 2022*). The experimental intervention was carried out three times a week for a period of six weeks, with a 48-hour rest between each session. Each session began with a normalized warm-up (10 min). The training intensity for both groups increased progressively on a weekly basis: during the 1st and 2nd weeks, they completed 15 sessions (3 times × 5 sets); in the 3rd and 4th weeks, they did 20 sessions (4 times × 5 sets); and in the 5th and 6th weeks, they performed 25 sessions (5 times × 5 sets). There was a 30-second rest between each repetition and a 5-minute break between sets. The training plan is shown in Table 2. In each repeated sprint, the two groups adopted a 15-second fixed duration mode, the instrument adopts a 3*3 (a total of nine points) site setting, and point 5 is set as the starting and ending point. Every time the subjects start from point 5 and sprint to any point with green light, they have to return to point 5 again to sprint next point with green light. Fixed sequence group complete 15-second repeated sprints in a fixed sequence according to 5-1-5-2-5-3-5-4......; Random sequence group complete 15-second repeated sprints in a random sequence according to 5- (1, 2, 3, 4, 6, 7, 8, 9 any point) -5- (1, 2, 3, 4, 6, 7, 8, 9 any point) -...... The training process is shown in Fig. 5. Upon completing the training sessions described, participants took a 10-minute break. Throughout the intervention, they were mandated to put forth their best effort in every session. The RPE scale was used for physiological monitoring, and the intensity reached grade 8–9.

## Statistical analysis

A two-way repeated measures analysis of variance (ANOVA) was carried out to evaluate the effect of the time (before and after intervention) and the effect of time-group interaction. The normality of distributions was verified using the Shapiro–Wilk test. Furthermore, Pearson correlation analyses were conducted to assess the relationships between T-drill, spider run, RA and TCH. Correlation coefficients were categorized as weak (0.1–0.3), moderate (0.4–0.7), and strong (>0.7) based on established criteria. All statistical analyses were conducted using SPSS22.

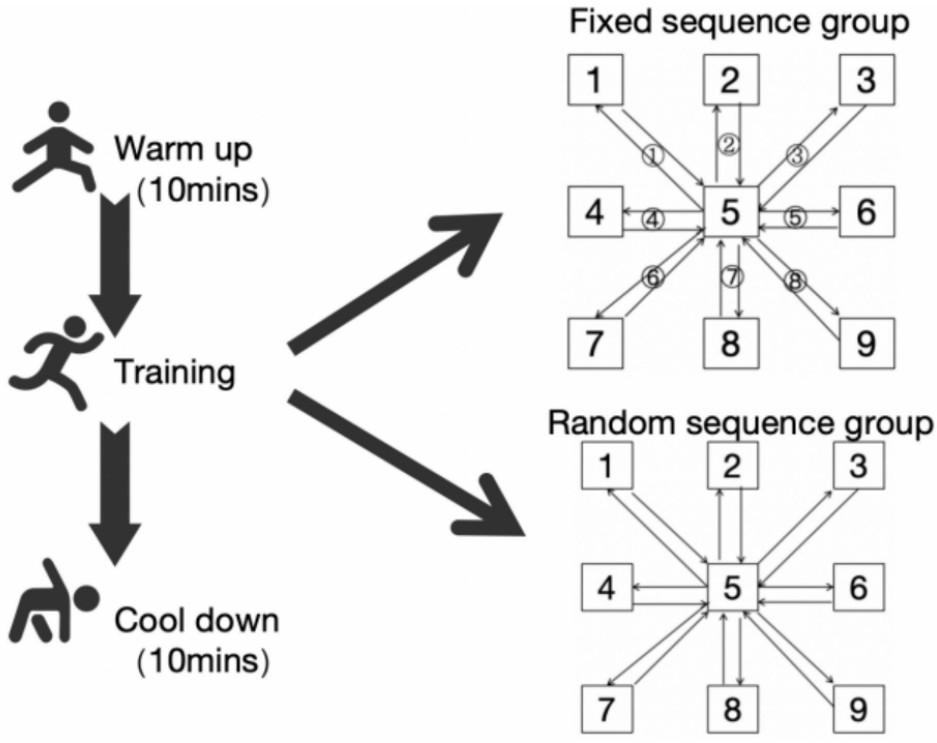

**Figure 5** Training flow diagram.

**Table 3** Effect of Speedcourt training on T-drill test.

| | Mean ± SD | Mean ± SD | $F$ | $p$ | $\eta 2$ |
|---|---|---|---|---|---|
| FS group | 8.74 ± 0.33 | 8.33 ± 0.32 | | | |
| RS group | 8.72 ± 0.35 | 8.20 ± 0.37 | | | |
| Group ME | | | 0.29 | 0.60 | 0.014 |
| Time ME | | | 349.47 | <0.001 | 0.94 |
| Group*Time | | | 4.10 | 0.057 | 0.17 |

Notes.
FS, fixed sequence; RS, random sequence; ME, main effect; $\eta 2$, partial $\eta 2$.

# RESULTS

## Effects of Speedcourt training on the T-drill test of female tennis players

As shown in Table 3 and Fig. 6, the two-way repeated measures results revealed a significant main effect of time on the T-drill performance ($F = 349.47$, $p < 0.001$, partial $\eta 2 = 0.94$). Moreover, there was no observed group main effect ($p = 0.60 > 0.05$) and group × time interaction effect ($p > 0.05$).

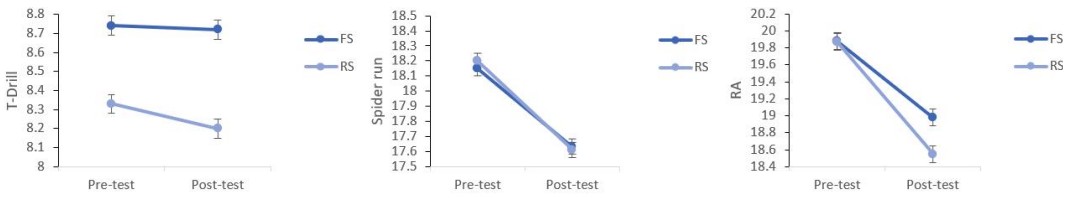

**Figure 6** Line graphs comparing pre- and post-test scores.

**Table 4** Effect of Speedcourt training on Spider run test.

| | Mean ± SD | Mean ± SD | $F$ | $p$ | $\eta2$ |
|---|---|---|---|---|---|
| FS group | 18.15 ± 0.59 | 17.63 ± 0.65 | | | |
| RS group | 18.20 ± 0.57 | 17.61 ± 0.63 | | | |
| Group ME | | | 0.003 | 0.96 | 0.01 |
| Time ME | | | 363.29 | <0.001 | 0.95 |
| Group*Time | | | 1.76 | 0.20 | 0.081 |

**Notes.**
FS, fixed sequence; RS, random sequence; ME, main effect; $\eta2$, partial $\eta2$.

### Effect of Speedcourt training on the spider run test of female tennis players

As presented in Table 4 and Fig. 6, the two-way repeated measures results highlighted a notably significant main effect of time on the spider run test ($F = 363.29$, $p < 0.001$, partial $\eta2 = 0.95$). Moreover, there was no observed group main effect ($p = 0.96 > 0.05$) and group × time interaction effect ($p > 0.05$).

### Effect of Speedcourt training on the reactive agility test of female tennis players

As shown in Table 5 and Fig. 6, the two-way repeated measures revealed a highly significant main effect of time on the RA test, with a large effect size ($F = 458.36$, $p < 0.001$, partial $\eta2 = 0.96$) Furthermore, there was a significant group × time interaction effect observed between the RS and FS groups, indicating different responses to the intervention over time ($F = 16.08$, $p = 0.001$, partial $\eta2 = 0.45$).

Due to the presence of time * group interaction effects, interaction and simple effects were further analyzed. The analysis of results showed that the simple effect of group was not significant in the pre-test ($p > 0.05$), In the post-test analysis, the simple effect of group was found to be statistically significant ($F = 4.50$, $p = 0.047 < 0.05$, partial $\eta2 = 0.184$).

### Correlation analysis

Table 6 displayed the correlations in three variables. The spider run test showed a strong correlation with the T-drill ($r = 0.790$), a moderate correlation with reactive agility (RA) ($r = 0.530$), and a moderate correlation with total court coverage (TCH) for both legs ($r = 0.654$, $r = 0.658$). The T-drill had a moderate correlation with RA ($r = 0.548$) and a moderate to strong correlation with TCH ($r = 0.683$, $r = 0.696$). Reactive agility was

**Table 5** Effect of Speedcourt training on RA.

| | Pre-test | Post-test | Repeated measures *F*-test | | |
|---|---|---|---|---|---|
| | Mean ± SD | Mean ± SD | *F* | *p* | $\eta 2$ |
| FS group | 19.88 ± 0.12 | 18.98 ± 0.14 | | | |
| RS group | 19.87 ± 0.11 | 18.55 ± 0.14 | | | |
| Group ME | | | 4.50 | 0.047 | 0.184 |
| Time ME | | | 458.36 | <0.001 | 0.96 |
| Group*Time | | | 16.08 | 0.001 | 0.45 |

**Notes.**

FS, fixed sequence; RS, random sequence; ME, main effect; $\eta 2$, partial $\eta 2$.

**Table 6** Correlation analysis.

| Index | SR | TD | RA | TCHR | TCHL |
|---|---|---|---|---|---|
| SR | 1 | | | | |
| TD | 0.790 | 1 | | | |
| RA | 0.530 | 0.548 | 1 | | |
| TCHR | 0.654 | 0.683 | 0.543 | 1 | |
| TCHL | 0.658 | 0.696 | 0.585 | 0.788 | 1 |

**Notes.**

SR, spider run; TD, T-drill; RA, reactive agility; TCHR, triple crossover hop (right); TCHL, triple crossover hop (left).

moderately correlated with TCH ($r = 0.543$, $r = 0.585$), and there was a strong correlation between the TCH of the right leg and the left leg ($r = 0.788$).

## DISCUSSION

The findings indicated that both groups experienced improvements in T-drill, spider run, and reactive agility (RA) following the intervention. When comparing the groups, the RS group demonstrated a more significant enhancement in RA compared to the FS group. Correlation analysis revealed that the spider run test had a strong correlation with the T-drill, and moderate correlations with RA and total court coverage (TCH). The T-drill also showed moderate correlations with RA and TCH. Additionally, RA exhibited a moderate correlation with TCH, and there was a strong correlation between the TCH of the right and left legs.

In this research, the T-drill was employed as a secondary measure to validate COD, while the spider run, a standard assessment tool for athletic COD within the International Tennis Number (ITN) test, was the primary measure of COD (*Zhou et al., 2024*). The outcomes indicated that both the spider run test and T-drill scores for the RS and FS groups increased significantly after the intervention compared to pre-intervention levels, with no significant differences between the two groups. *Born et al. (2016)* found Speedcourt sprint training can significantly improve the Illinois test in elite soccer player. Another study *Liu (2019)* reported significant enhancements in COD test performance following eight weeks of multi-directional training. Furthermore, a separate study noted improvements in COD speed for both groups after three weeks of engaging in multi-directional sprinting, utilizing both random and fixed routes (*Zhou et al., 2024*). Consistent with the results of this study,

the Speedcourt training improved the COD speed of the subjects, and the enhancement in COD performance may be attributed to neural adaptations and improvements in motor unit recruitment (*Pardos-Mainer et al., 2021*). The central nervous system of the subjects developed adaptations to the rapid-onset change-of-direction pattern (*Raeder et al., 2024*). In addition, the improvement effect observed in the study did not show any significant differences between the two groups, as neither the T-drill nor the spider run test involved decision-making factors (*Sagat et al., 2023*), and COD tasks are typically closed skills involving pre-planned movements, which may contribute to the similarity in improvement across both groups (*Dos'Santos et al., 2017*).

The study showed that both groups experienced significant improvements in RA following the exercise intervention, with the RS group exhibiting a more pronounced enhancement in RA compared to the FS group. Although no previous studies have explored the effects of Speedcourt training on the RA of female tennis players, studies have found that after elite female soccer players perform agility sprints with different schemes based on Speedcourt equipment, the reactive time of subjects has been improved (*Raeder et al., 2024*). *Chaouachi et al. (2014)* found significant improvements in sprint, COD, and reactive agility in football players through 6 weeks of multi-directional sprint training. It can be seen that the current findings in this study demonstrate congruence with prior investigations in this domain. The enhancement of RA primarily hinged on improvements in perception and reaction time to specific external stimuli, rather than on the actual speed of movement (*Young & Rogers, 2014*). RA is traditionally considered an integral component of physical performance. However, emerging evidence suggests that RA may be more strongly influenced by cognitive parameters than by physical attributes (*Scanlan et al., 2014*). Specifically, RA demonstrates a higher correlation with response time and decision-making time—both cognitive metrics—compared to traditional physical performance measures such as sprint speed, COD speed, or other physical parameters (*Scanlan et al., 2014*). Based on these factors recent research suggested combining sport-relevant motor and cognitive functions in dual-task approaches to enhance ecological validity and increase transfer effects (*Scharfen & Memmert, 2021*).

The correlation analysis results of this study indicated that the spider run test had a strong correlation with the T-drill ($r = 0.790$), and moderate correlations with reactive agility (RA) ($r = 0.530$) and total court coverage (TCH) for both legs ($r = 0.654$, $r = 0.658$). The T-drill showed moderate correlations with RA ($r = 0.548$) and TCH ($r = 0.683$, $r = 0.696$). Additionally, RA had moderate correlations with TCH ($r = 0.543$, $r = 0.585$), and there was a strong correlation between the TCH of the right leg and the left leg ($r = 0.788$). Previous studies (*Condello et al., 2013*; *Falces-Prieto et al., 2022*) have shown a moderate correlation between TCH and COD, and a moderate correlation between RA and pre-planned COD. The study believes that both the spider run and T-drill are pre-planned COD tests, so there is a strong correlation between them, while the RA test contains more complex situations: visual perception, body movements control in space, coordination and so on. It is the cause that between RA and COD were only moderately correlated. A study has suggested that unilateral exercises may impose demands akin to those found in every COD maneuver (*Suarez-Arrones et al., 2020*). Consequently, it is plausible that

unilateral actions, such as unilateral jumps or power movements (horizontal, lateral, or vertical), may exhibit stronger correlations with COD. This aligns with the findings of the current study. For effective change of direction, an acceleration and propulsion phase are necessary, which involves ground contact times longer than 250 ms and significant angular displacement between joints. It is thus anticipated that the long stretch-shortening cycle plays a more decisive role in COD. As the TCH represents a long stretch-shortening cycle action that necessitates time to generate force for propulsion, strong associations between COD and TCH are expected (*Vescovi & Mcguigan, 2008*). Studies have demonstrated that the asymmetry in bilateral lower limb strength is correlated with COD ability to varying degrees, and this correlation also exhibits gender differences (*Bishop et al., 2021*; *Ascenzi et al., 2020*). Associations between the asymmetry in bilateral lower limb strength and COD performance were moderate in male tennis players, but for female athletes, the associations were almost exclusively statisticfally non-significant. Compared with male athletes, female athletes rely more heavily on symmetrical bilateral strength to enhance COD efficiency. Therefore, there is no difference in the correlation between the dominant and non-dominant legs and COD performance for female athletes (*Kozinc & Šarabon, 2024*).

## CONCLUSION

Six-week Speedcourt training can effectively enhance the agility and change-of-direction of female tennis players. Also, incorporating lower limb explosiveness exercises in agility-specific training may have a greater improving effect on the agility of female tennis players.

## ACKNOWLEDGEMENTS

The authors thank all the subjects for their efforts, and the authors declare no conflict of interest.

### Funding

The authors received no funding for this work.

### Competing Interests

The authors declare there are no competing interests.

### Author Contributions

- Zhihui Zhou conceived and designed the experiments, performed the experiments, analyzed the data, prepared figures and/or tables, authored or reviewed drafts of the article, and approved the final draft.
- Jiawei Wang performed the experiments, authored or reviewed drafts of the article, and approved the final draft.
- Hao Wang analyzed the data, prepared figures and/or tables, authored or reviewed drafts of the article, and approved the final draft.

- Guo Ru analyzed the data, prepared figures and/or tables, and approved the final draft.
- Fanhui Kong conceived and designed the experiments, analyzed the data, authored or reviewed drafts of the article, and approved the final draft.

### Human Ethics

The following information was supplied relating to ethical approvals (*i.e.*, approving body and any reference numbers):

The Ethics Committee of Beijing Sport University (2024202H).

### Data Availability

The raw measurements are available in the Supplementary File.

### Supplemental Information

Supplemental information for this article can be found online at http://dx.doi.org/10.7717/peerj.19339#supplemental-information.

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
