# Peer review of "A study of female tennis players: Speedcourt training is effective on improving agility and change-of-direction"

_PeerJ, doi:10.7717/peerj.19339_

## Round 0.1 · original submission · Major Revisions

Dear Authors

Your study has been reviewed by two experts in the study field. The comprehensive comments to improve the quality of the manuscript have been addressed. I would encourage you to highlight the points raised by the reviewers, particular the comment related to no control group to justify the baseline measures. We invite you to submit a revised version of the manuscript addressing the reviewers’ comments.

We look forward to receiving your revised manuscript.

Best regards

Yung-Sheng Chen, Ph.D.
Academic Editor

·

Basic reporting

The study examines the effects of Speedcourt training on COD, agility and reactive agility skills of female tennis athletes. In terms of the subject of the research article, although it sheds light on the deficiency in the literature, certain deficiencies stand out.
Introduction: The last paragraphs of the research paper, especially the introduction, should clearly emphasise the lack of emphasis in the literature.
Method: It would be more understandable if the training programme was given in a table. It is also seen that no measurement was made on the speedcourt equipment after the test protocols. When I read the title of the article, I was under the impression that a test was performed on the speedcourt. At this point, I wonder if the title could be revised. At this point, I think it would be more accurate to write a statement on agility and change of direction in the title.
Discussion: Generally, in the literature, comparisons were made with the data obtained from the results of male or female football players. If there is a lack of research on female athletes, it should be stated. In addition, there are studies in the literature examining the relationship between dominant and nondominate leg strength levels and COD performance. At this point, I suggest that the discussion section should be revised again.
Apart from these, if there are Speedcourt tests, I recommend you to add them as well.
Thank you for your endeavours for your research article and I wish you success. have a good study.

Experimental design

The research looked for more effects on change of direction running ability. In this section, there is only one test measuring reactive agility that looks at change of direction running after agility training. I suggest adding the Speedcourt test to make the research paper more logical and strengthen it.

Validity of the findings

no comment

Additional comments

In general, the article writing could be better. I think there are also some language conflicts. The title and content should be revised and literature examples should be increased if possible. The training part should be presented in tables and the relationship between right and left leg strength tests and agility skills should be given in the introduction part with literature examples and should be linked to a purpose.

Reviewer 2 ·

Basic reporting

General Comments; Scientific Implications of the Missing Control Group
1. Potential Confounding Variables: Without a control group, it is difficult to differentiate between the true effect of the intervention and other external influences such as natural physiological adaptations, prior agility training, or concurrent physical activities.
2. Lack of Baseline Comparison for Training-Induced Gains: A non-intervention control group would provide a critical baseline, ensuring that the improvements observed in the RS and FS groups were not simply due to normal training progression or placebo effects.
3. Threats to Internal Validity: The absence of a control condition limits causal inferences, making it unclear whether Speedcourt training alone was responsible for agility enhancements or if other unaccounted factors contributed.

Experimental design

Materials and Methods
• Clearly defines inclusion/exclusion criteria (Lines 73-78).
• Ethical approval is properly mentioned (Lines 79-80).
Weaknesses & Suggested Improvements
No Mention of Prior Agility Training (Lines 74-76)
a. The study excludes players with injuries but does not mention previous agility training exposure.
Add; Clarify whether participants had previous agility-specific training.
Randomization Process Not Specified (Lines 69-70)
a. The study states that players were randomly assigned, but the method is unclear.
Add; Specify whether computer-generated randomization or block randomization was used.
RA Test Execution Not Clearly Explained (Lines 104-107)
a. The RA test includes visual stimuli, but how reaction time is measured is unclear.
Specify whether reaction time was recorded via Speedcourt software or manually.
No Citation for TCH Test Reliability (Lines 111-117)
a. The TCH test lacks a reference supporting its reliability for unilateral explosive power.
Add a citation verifying TCH as a valid explosive power measure
No Justification for Six-Week Training Duration (Lines 126-127)
a. Cite previous studies demonstrating why six weeks is optimal for agility development.
Training Load Monitoring Not Mentioned (Lines 130-134)
a. Explain if physiological measures (heart rate, RPE) were used to monitor training intensity.

Validity of the findings

Results (Lines 149-174)
• Clearly reports statistically significant findings (Lines 152-164).
• Correlation analysis provides meaningful insights (Lines 169-174).
Weaknesses & Suggested Improvements
No Graphical Representation of Data (Lines 157-159)
a. Include line graphs comparing pre- and post-test scores.

Additional comments

One of the most significant methodological limitations of this study is the absence of a control group. While the inclusion of two experimental groups—Random Sequence (RS) and Fixed Sequence (FS) Speedcourt training groups—allows for a comparative analysis of training variations, the study does not include a non-intervention control group. This omission raises concerns regarding the extent to which the reported agility improvements can be attributed solely to the Speedcourt training intervention.

Annotated reviews are not available for download in order to protect the identity of reviewers who chose to remain anonymous.

---

## Round 0.2 · Minor Revisions

Dear Authors

Your revision and point-by-point responses to reviewers' comments have been assessed. However, we believe a minor revision is required to improve the quality of the manuscript in its current form. Please view the attachment of the annotated manuscript as the points raised by the reviewers. We invite you to submit a revised version of the manuscript addressing the reviewers’ comments.

We look forward to receiving your revised manuscript.

Best regards

Yung-Sheng Chen, Ph.D.
Academic Editor

·

Basic reporting

The requested corrections have improved the research paper. However, minor corrections are necessary. The last part of the Objective section in the Abstract should be revised. Is the only deficiency the small number of studies? In this section, it would be good to emphasise that the number of exercises such as speedcourt is also small.
In the same section, a little information about unilateral measurements should be added in the Method section.
In the Introduction section, lines 41 and 42 should be reviewed whether it is COD or agility skill.
In lines 57 and 58, a reference to ‘It can design different tests and training programmes for different sports’ should be added.
64 and 65 lines ‘However, in the existing research, compared with the male group, there are few studies on female tennis players’ should be revised.
77 line= The purpose of the research should be added after the hypothesis.
95 -142 lines= Details of the warm-up phases should be written.
231 lines= ‘However, emerging evidence suggests that RA may be more strongly influenced by cognitive parameters than by physical attributes’ reference should be added.

Experimental design

Details of the warming phases should be given.

Validity of the findings

no comment

Additional comments

The requested corrections have improved the research paper. However, minor corrections are necessary. The last part of the Objective section in the Abstract should be revised. Is the only deficiency the small number of studies? In this section, it would be good to emphasise that the number of exercises such as speedcourt is also small.
In the same section, a little information about unilateral measurements should be added in the Method section.
In the Introduction section, lines 41 and 42 should be reviewed whether it is COD or agility skill.
In lines 57 and 58, a reference to ‘It can design different tests and training programmes for different sports’ should be added.
64 and 65 lines ‘However, in the existing research, compared with the male group, there are few studies on female tennis players’ should be revised.
77 line= The purpose of the research should be added after the hypothesis.
95 -142 lines= Details of the warm-up phases should be written.
231 lines= ‘However, emerging evidence suggests that RA may be more strongly influenced by cognitive parameters than by physical attributes’ reference should be added.

Reviewer 2 ·

Basic reporting

The manuscript is now written in clear and professional English, with previous issues related to grammar and expression corrected. The literature review is sufficient and provides a clear context for the study. Additionally, the inclusion of numerical results, effect sizes, and visual data representations, as previously suggested, has improved the clarity and academic quality of the results section.

Experimental design

Although the lack of a control group remains a methodological limitation, the authors have now provided a clearer rationale for the research hypothesis and the selection of the Speedcourt training method. The previous suggestions regarding randomization procedures, test execution details, and justification for the training duration have been appropriately addressed, resulting in a more robust methodological design.

Validity of the findings

The findings are statistically sound and the improvements made to the manuscript have strengthened the generalizability of the results. The correlation analyses and the presentation of results are now more detailed and transparent, as suggested in the initial review. However, due to the absence of a control group, internal validity is still somewhat limited, and conclusions should be interpreted in the context of comparisons between the two experimental conditions only.

Additional comments

The authors have carefully addressed all suggestions made in the previous review, leading to a significantly improved manuscript. In particular, the addition of graphs and clearer result descriptions enhance the readability and impact of the study. For future research, the inclusion of a control group and a longer training duration are recommended to further strengthen the scientific value and reliability of the findings.

Final recommendation: The manuscript is now acceptable in its current form.

Annotated reviews are not available for download in order to protect the identity of reviewers who chose to remain anonymous.

---

## Round 0.3 · accepted · Accept

Dear Authors,

I would like to express my big heart for your patience and efforts to improve the quality of the manuscript. Two experts have now endorsed your submission for acceptance of publication in PeerJ. Congratulation!!!

Thank you for submitting your article to PeerJ. I look forward to receiving your research and review articles in the future.

Best Regards
Ph.D. Yung-Sheng Chen

·

Basic reporting

Good

Experimental design

Good

Validity of the findings

Good

Additional comments

Good